# Interplay of Structural Factors in Formation of Microphase-Separated or Microphase-Mixed Structures of Polyurethanes Revealed by Solid-State NMR and Dielectric Spectroscopy

**DOI:** 10.3390/polym13121967

**Published:** 2021-06-14

**Authors:** Stepan A. Ostanin, Maxim V. Mokeev, Dmitry V. Pikhurov, Aleksandr S. Sakhatskii, Vjacheslav V. Zuev

**Affiliations:** 1ITMO University, Kronverkskiy pr. 49, 197101 Saint-Petersburg, Russia; stepan.ostanin1995@gmail.com; 2Institute of Macromolecular Compounds of the Russian Academy of Sciences, Bolshoi pr. 31, 199004 Saint Petersburg, Russia; linxup@gmail.com; 3Varnish-Paint Plant “CRONOS SPb”, Polevaya Sabirovskaya 42, 197183 Saint Petersburg, Russia; nefaeron@gmail.com; 4Saint-Petersburg State University, 7/9 Universitetskaya nab., 199034 Saint Petersburg, Russia; ale.x-man@mail.ru

**Keywords:** solid-state NMR, FTIR, polyurethane, domain size, microphase separation, dielectric spectroscopy

## Abstract

A set of aromatic-oxyaliphatic polyurethanes (PUs) with different mass fractions of components also containing fluorinated fragments was synthesized and studied using various solid-state NMR techniques and dielectric spectroscopy. In contrast to the common model suggested by Cooper and Tobolsky in 1966, the rigid domains of microphase separated PUs are formed, not only by units containing urethane bonds, but also by oxyethylene fragments that form a common rigid phase. The urethane bonds and oxyethylene fragments are incorporated into both rigid and soft phases. Good agreement with the Cooper and Tobolsky model is observed only when solubility parameters are significantly different for the hard and soft segments, such as hydrocarbon aromatics and perfluoroaliphatic blocks.

## 1. Introduction

Polyurethanes (PU) are randomly segmented copolymers consisting of alternating flexible soft segments (SS) and formal rigid urethane-containing hard segments (HS) [1,2]. Their properties are directly related to their hierarchical morphology and strongly depend on the degree of microphase separation. This degree, in turn, is a function of several parameters, such as the system thermodynamics, the ability of HS to pack correctly to form hydrogen bonds, the chemical structure of copolymers and their composition, synthesis conditions, and thermal history, etc. [3]. The conventional wisdom is that HSs tend to cluster or aggregate into ordered nanosized hard domains [4] as a result of the formation of hydrogen bonds between carbonyl to amino functions, which are in urethane groups, whereas SSs form an amorphous continuous matrix. The HSs acts as filler particles as well as crosslinkers restraining the mobility of SS chains. Such structure was first proposed by Cooper and Tobolsky in 1966 [5]. The degree to which the hard and soft phases separate plays a vital role in determining the bulk properties of these multi-block copolymers. Important factors that affect this phase separation are the hard/soft segment ratio in the copolymer, the crystallizability of hard and soft segments, the extent of competitive hydrogen bonding between hard–hard and hard–soft segments, and the mutual solubility between hard and soft segments [6,7]. All these parameters are connected to molecular mobility, which determines a number of polymer properties, including mechanical and thermoplastic characteristics. With so many variables to consider, investigation of the morphological behavior of PUs and interpretation of their structure–morphology–property behavior has been a very active but also a very challenging field of research [8].

PUs often show multiple endothermic signals in DSC thermograms during heating, which are commonly interpreted as melting, and the multiple peaks are interpreted in terms of the phase behavior of these PUs [9]. This interpretation contradicts the simple Cooper–Tobolsky model and structural studies. However, it is clear that the reorganization of PU polymer matrix upon heating can be related to the formation or breakdown of H-bonds (inter- or intra-chain), and such processes can be cooperative. These changes are reflected in the behavior of certain parameters, such as the domain size, interdomain distance, domain density, and their volume fraction, as well as the degree of microphase separation. Because of the modest scattering contrast between the rigid domains and soft matrix, it is difficult to observe the phase structure by scattering methods [10]. This is why a number of various solid-state NMR techniques have been applied to study PU structures, including domain size and interdomain distance elucidation [11]. It is generally accepted that, in terms of molecular mobility, the soft matrix contains two fractions: rigid and mobile (soft) [12]. Many authors have proposed models with space separation of phases of the soft matrix with distinct borders between zones with different mobility corresponding to the presence of different phases [13]. However, the dielectric spectroscopy of PUs shows that the soft matrix has only one α- relaxation process (one glass transition), and hence constitutes a single phase [14].

This investigation has several major objectives: the first one is to study the effect of the chemical structure of PU building blocks on H-bonding and microphase separation. The N–H^…^O=C H- bond between two urethane groups is commonly considered as the strongest one. However, some studies have shown that the N–H^…^O (ether) H-bond is slightly stronger than the N–H^…^O=C interchain bond [15]. On the other hand, it is known that the energy of a hydrogen bond depends on the length and nature of the soft segments and the concentration of the hard segments in a chain [16]. Competition between urethane and ether oxygen atoms in forming hydrogen bonds should lead to a more brittle structure of rigid domains in the case of PUs with ethylene oxide based soft blocks. For this purpose, we designed and synthesized PUs with a minimal number of structural variables. As the soft segment we used PEG-600 and fluorinated diol of the same length. This allows one to compare the effect of the solubility parameter of the segments and their polarizability on the structure and molecular mobility in PUs. Another structural motif was to change the soft segment to an aromatic fragment with or without a fluorinated part with the same motivation. This allows one to increase the volume fraction of HS and to introduce disorder in rigid domains. We used various solid-state NMR techniques, BDS dielectric spectroscopy, and FTIR spectroscopy in order to characterize the structure and dynamics of the systems studied.

## 2. Experimental Part

### 2.1. Materials

The isocyanate used in this study was an oligomeric aromatic isocyanate based on 4,4′-diisocyanate diphenylmethane (MDI) Wannate PM-200 from Yantai Wanhua (PR, China) (molecular weight M_n_ = 340, 30.2–32% NCO, viscosity at 25 °C 150–250 cps, specific gravity d = 1.25 g/cc). As a polyethylene oxide containing polyol, PEG 600 from LOBA Chemie with M_n_ = 600 and specific gravity 1.01 g/cc was used. Fluorinated diol Fluorolink E10-H with M_n_ = 1700 and specific gravity 1.69 g/cc was obtained from Solvay Solexis. Bisphenol A was obtained from Reachim (Russia). α, α, α, α’, α’, α’-hexafluorobisphenol A (T_m_ = 160 °C) was obtained from AlfaFtor (Russia).

All samples were prepared using the prepolymer technique [1] (Scheme 1). The formulations were calculated so that the molar ratio of NCO and OH groups was 1:1.

### 2.2. Methods

NMR experiments were performed on a Bruker Avance II 500 spectrometer operating at a proton frequency of 500.13 MHz. Four millimeter and seven millimeter double resonance MAS probe heads were used for recording ^13^C spectra under MAS conditions and ^1^H static spin-diffusion experiments, respectively. We used the methods described in papers [17,18,19,20,21,22,23,24] (see Appendix A).

The FTIR absorbance spectra of PU thin films were recorded using a Bruker Vertex 70 spectrometer.

Dielectric measurements were performed as described previously (see Appendix A).

## 3. Results and Discussion

### 3.1. Synthesis of PUs

To investigate, at a molecular level, the role of structural factors on the formation of the microstructure and in controlling the dynamics of segmented polyurethanes (PUs), a series of copolymers was systematically designed and synthesized. The motivation for their design was the following: the existing morphology in PU materials is determined by a balance between competing forces that involve intermolecular interactions and entropic degrees of freedom, which are connected to molecular dynamics [25]. The thermodynamic driving forces lead to the equilibrium morphology of these multi-block copolymers in the bulk state, which depends on the block volume fraction (f) and the Flory−Huggins interaction parameter (χ) [26]. The attainment of the equilibrium morphology also depends on the molecular mobility in the bulk state. The block volume fractions can be easily varied by changing the chain extender fraction. The variation of the Flory−Huggins interaction parameters, while keeping other properties of the polymer systems under study fixed, can be realized by the substitution of hydrogen atoms by fluorine. The fluorine has practically the same volume as hydrogen but differs in polarizability and polarity of the C-F bond [27].

As the object of this study, we choose the industrial type PU system with oligomeric 4,4′-diphenylmethane diisocyanate as the isocyanate component (the cheapest commercially available), bisphenol A or its fluorinated analogues as the chain extender, and PEG-600 or its fluorinated analogues as the soft segments. The choice of PEG-600 as a soft fragment was motivated by the fact that, in PUs, two types of hydrogen bonds can be formed between NH groups and proton accepting oxygen in urethane C=O groups and ether C–O–C groups. The types and strengths of the hydrogen bonds are usually estimated by the magnitude of the wave number shifts in the IR spectrum. On the other hand, the polyethylene oxide fragments have a tendency to crystallize, which can lead to formation of an “inverse” structure of PUs, where hard domains are formed by crystallites of polyethylene oxide fragments, and urethane containing units are located in the soft matrix.

Based on this motivation, we synthesized a number of PUs with varied volume fractions of fluorinated and non-fluorinated soft and hard segments. Bisphenol A, or its fluorinated analogue, were used as chain extenders. The compositions of all samples under study are given in Table 1. All samples were loosely sewn networks, which did not prevent microphase separation but would not allow determination of their molecular weights. The advantage of this approach is based on the possibility to eliminate the effect of the molecular mass of different samples on the study of microphase separation. The use of dual-NCO-functionalized MDI in combination with polyols with different reactivity inevitably leads to formation of polymers with very different molecular masses. However, with our approach, all samples form loosely sewn networks with a similar effect on microphase separation.

In view of this background, we attempted to systematically understand the role of microstructural design in control of the dynamic mobility of various polyurethane microstructures by focusing on the chemical structures in these systems.

### 3.2. Solid-State NMR Experiments

We employed advanced solid-state NMR spectroscopic methods to study the chemical structures of different phases of PUs with different molecular mobility in order to understand the effect of chemical structure on microphase separation. In addition, insights into molecular ordering of the HS and SS in the different structural modifications are gained to reveal an interplay and interdependencies of the packing tendencies of the rigid backbone, flexible chains, and hydrogen bonding between and into different type of segments. The common model of PUs suggested by Cooper and Tobolsky is that their organization as a mixture of rigid domains dissolved in an amorphous matrix [5]. The rigid domains of PUs are close analogues of crystallites. These rigid domains have a strong network of hydrogen bonds and are in most cases infusible [4] and show restricted molecular mobility. In addition, amorphous regions with high glass transition temperatures [28] and restricted mobility, according to NMR data [29], can exist in the interior of these domains. However, polymer chains in these latter regions are more mobile than in the other parts of rigid domains. The amorphous polymer matrix has a complex landscape of mobilities according to solid-state NMR data [19]. The commonly accepted terminology makes use of terms such as rigid and mobile amorphous phase. However, it has to be appreciated that it is not a phase in a true thermodynamic sense, because amorphous phases have a single glass transition. These terms are used to emphasize the presence of polymer chains or their segments with different mobility, which can be located in aggregates but do not form a true phase. One of the possible ways of their formation is by H-bonding. It has been known for more than 50 years that crosslinks put constant restrictions on polymer segmental motions [30]. The formation of H-bonds in a soft amorphous matrix is analogous to the formation of knots in a network. Hence, from the point of view of mobility studied by NMR, this leads to the formation of spatial regions with restricted molecular mobility near these knots (H-bonds), which we define as rigid amorphous phase, and other parts of the amorphous matrix without these restrictions (soft phase).

^13^C CP/MAS experiments provide information about the chemical composition of regions with restricted mobility, e.g., in our case about the structure of the rigid domains or amorphous phase, if rigid domains are not formed. The last case corresponds to the formation of amorphous microphase-mixed structures with restricted mobility (rigid amorphous phase). However, it is difficult to make an accurate distinction between these two structures (microphase-mixed and microphase-separated) based only on this type of solid-state NMR experiment. CP/MAS experiments with a short contact time (100 µs) and high power proton decoupling are more suitable for the rigid part of multiphase materials, because cross polarization has a low efficiency for the mobile phase due to high mobility. We applied these conditions in our experiments. (see Appendix A). Direct polarization MAS experiment with low power proton decoupling gives signals only from the soft phase. Echo detection in such experiments provides additional filtering of the rigid component. CPSP experiments combine CP efficiency for the rigid phase and direct polarization of the mobile phase in one experiment.

First, we studied the sample **R_1_/100** (with 100 mol.% of PEG-600 as diol component). Figure 1(a1) presents the ^13^C CP/MAS spectrum of this sample. The rigid phase contains both ethylene oxide and aromatic fragments. Hence, this result supports the formation of a rigid low mobile phase through both N–H^…^O (ether) and N–H^…^O=C (urethane) hydrogen bonds. To observe the signals from both rigid and mobile phases, we recorded the ^13^C CPSP/MAS spectrum [29]. In this spectrum (Figure 1(a2)) one can see the appearance of additional signals of terminal fragments of ethylene oxide units. To obtain the spectrum of the mobile phase in these samples, we used a direct-polarization MAS NMR experiment with echo detection [31]. In this case (Figure 1(a3)), the mobile phase contains only signals of ethylene oxide and methylene groups of diphenylmethane fragments. It is important to note that the rigid phase with restricted mobility practically did not contain terminal fragments of ethylene oxide units, and, as seen from the spectra of the **R_1_/100**-sample normalized with respect to the signal intensity of the ethylene oxide (the integral intensity of this line is the same in all spectra) (Figure 1a), the mobile phase is enriched by terminal fragments of ethylene oxide units. The assignment of signals to terminal groups is based on solution NMR spectra of starting polyol. Hence, we can suggest that rigid domains are built from hydrogen bonded through both N–H^…^O (ether) and N–H^…^O=C (urethane) units containing ethylene oxide and diphenylmethane fragments. An alternative explanation is that crystallization of PEG-600 fragments possibly takes place. The melting point of pure PEG-600 is 17–22 °C. Upon introduction of PEG fragments into the PUs structure, the possibility of their crystallization appears to depend on the isocyanate used, and start only with a molecular weight of PEG higher than 1000 [32]. We recorded DSC scans for the PU samples with PEG-600 fragments (see Appendix A). As one can see, all DSC curves did not show peaks that could be attributed to the melting of PEG fragments. Hence, the restricted mobility of the ethylene oxide units is determined by their interactions with urethane units (H-bonds formation).

In order to investigate the effect of the aromatic chain extender on the microphase structure of PUs, we synthesized a number of samples in which a part of PEG-600 was substituted to a bisphenol A (Samples 2–5). We performed the same experiments as with the sample **R_1_/100**. As one can see from Figure 1c, the rigid phase contains all the units used at synthesis. Hence, the presence of ethylene oxide in all cases leads to its presence in the rigid domains, and these rigid domains have complex structure incorporating all units used at the synthesis of PUs. However, the mobile phase is enriched by the bisphenol A fragments. When the molar fraction of bisphenol A was increased up to 80% relative to the PEG-600, the mobile phase disappeared from the direct-polarization MAS NMR spectrum (Figure 1b). Hence, the distribution of molecular mobility in the polymer volume is determined, not only by their phase structure (rigid domains vs. amorphous polymer matrix), but also by the character and geometry of the hydrogen bonds, which depend on the molecular structure of polymer segments (presence of kinks due to the presence of bisphenol A or oligomeric MDI units, and length ratios of molecular segments). The effect of the length of oligomeric MDI units is clearly observed in the ^13^C CP/MAS spectra of **R_1_/100** (Figure 1a). Oligomeric MDI is a mixture of two, three, or more aromatic rings containing isocyanates. As one can see, the rigid phases are enriched by long units of oligomeric MDI in contrast to the mobile phase, which is enriched by the short ones.

We synthesized a sample **R_3_/100**, in which PEG-600 was fully substituted to the bisphenol A. As expected, this sample, did not contain a mobile phase (see Appendix A), and the ^13^C CP/MAS and ^13^C CPSP/MAS spectra completely coincided. The same result was obtained when bisphenol A was replaced by its fluorinated analog (see Appendix A). However, the spectral data indicate the presence of two rigid phases, namely, rigid domains and a rigid amorphous phase, the relative fraction each varying significantly from sample to sample. Hence, the polymer **R_3_/100** has a typical biphasic structure of semictystalline polymers (amorphous/crystalline).

The next sample under study was **R_2_/100**, in which PEG-600 was substituted for its perfluorinated analog (Figure 1d). As one can see from the spectra, the rigid phase contains only aromatic-urethane units, and the mobile phase contains only fluorinated units. Hence, a full microphase separation of isocyanate and polyol components, which was predicted for PUs [5], is achieved only in the case of strongly different Flory−Huggins interaction parameters. A substitution of 50 mol.% of perfluorinated diol to the bisphenol A leads to a PU in which the rigid phase is very diffuse and has the character of an interphase that includes terminal groups around urethane bonds. All PUs with a 1:1 ratio of two diol components reveal a specific behavior, which may be attributed to the formation of a eutectic system. Thus, the PU in which 50 mol.% of PEG-600 was substituted on fluorinated analog of bisphenol A did not give any signal in the direct-polarization MAS NMR spectrum, i.e., it did not contain any mobile phase (see Appendix A).

It is interesting to compare our results with the results of IR spectroscopy, which is usually used to estimate the phase separation of PUs [33]. This method was descripted in detail in our previous paper [26] (see Appendix A). Selected IR spectra of the PUs under study in the carbonyl stretching region with deconvolution are given in Figure 2, and the values of degree of phase separation, λ_DPS_, are given in Table 1. The current concept of microphase separation in PUs assumes that all hydrogen bonded urethane units are located in rigid domains, while the free urethane units are located in the soft matrix [31]. Our NMR results show that urethane containing units are located only in low mobile rigid phases, i.e., in both rigid domains and rigid amorphous phases. Hence, the numbers obtained by means of IR spectrosopy do not measure the degree of microphase separation, but rather the degree of ordering of hydrogen bonds. The introduction of composition disordering in PUs structure (Samples 2–5, introduction of aromatic chain extender, which is a kink that prevents linear organization of polymer chains or segments, and, therefore, the formation of interchain hydrogen bonds) leads to an increase of disorder, in both the rigid domains and the amorphous phase, and, therefore, to a decrease of λ_DPS_. Hence, the deviation of λ_DPS_ values from unity confirms the difference in the structure of rigid domains and polymer crystallites. This suggests that the structure of the rigid domains is similar to the rigid amorphous phases, such as the one in a tightly sewn epoxy resin [32], but they differ in the density of H-bonds and their geometry, which leads to a difference in molecular mobility. Hence, the word “rigid”, when applied to the rigid domains and rigid amorphous phases, means a restricted mobility relatively to the soft phases, but relative to others, the rigid amorphous phase is soft, which is clear from the results of the T_2_ relaxation measurements (see Section 3.3.1, ^1^H Transverse Magnetization Relaxation).

#### The Size of Rigid Domains

The presence of rigid domains in microphase-separated PUs governs their physical properties. Hence, estimating the domain sizes and structural (and other) factors that determine their dimensions is an important task with regard to the possibility of producing novel PU materials with a high potential for new applications. In order to determine quantitatively the size of rigid domains, we used the proton spin diffusion NMR technique [33]. Domain sizes are evaluated via spin diffusion NMR by measuring the rates of change of nuclear spin polarizations associated with the two phases in a PU. When a step-function initial polarization gradient is imposed on a two-component system, the initial rate of exchange is directly proportional to the inter-component surface-area/volume ratio in the sample. From that ratio, the domain size can be calculated because the cylindrical shape of rigid domains in PUs is known [26,34]. As previously, we used proton spin-diffusion experiments with a double-quantum (DQ) filter [35]. The precision of ^1^H spin diffusion NMR experiments, as well as straightforward experimental protocols, depend on the mobility properties of the systems under study. We tested it by studying DQ buildup curves (Figure 3). The DQ buildup curves were obtained by employing a pulse sequence suggested in [18] (see Appendix A). As expected, for the systems with rigid domains and a more or less mobile amorphous phase, a superposition of two curves was observed because the DQ dipolar filter is highly sensitive to the heterogeneity of the dipolar network [36]. The maxima of the curves appear at very short excitation time, t, in a very narrow range from 18 to 20 µs, indicating the presence of strong ^1^H dipolar interactions. These interactions are stronger in the sample with perfluorinated SS than in other samples, which is consistent with stronger separation of HS and SS blocks. The ^19^F effects on the ^1^H spin diffusion are not expected to make a considerable contribution. This is based on the observation that the presence of fluorinated fragments did not influence solid-state proton spectra (see Appendix A).

The domain sizes were calculated by utilizing a method suggested in [22] and used by us previously [26,35]. The parameters used for the calculation (see Appendix A) and the results are given in Table 2. The rigid domains in PUs under study have cylindrical shape that was supported by SEM microscopy (see Appendix A). Examples of the spin-diffusion curves used for estimation of the square root of the mixing times, t_m_, used in the calculation of domain sizes are given in Figure 4.

The dimensions of the rigid domains in PUs with ethylene oxide SS are very similar to those previously reported by us for comparable structures (the same isocyanate component with variation of SS) [35]. An increase of the volume fraction of HS in PUs leads to an increase of the size of rigid domains, concurrently with the interdomain distances, because the number of domains remains approximately constant [36]. Interestingly, the smallest dimensions in this series are observed for PUs with an equal ratio of PEG-600 and bisphenol A, similar to the formation of a eutectic system [37].

The PU with perfluorinated SS produced rigid domains that were anomalously small in size. This can be explained by the low volume fraction of HS and by the incompatibility of perfluorinated SS with HS. However, as one can see from the spacing between the rigid domains, their number (number per unit volume) is similar to those observed with nonfluorinated analogues.

The effect of the incompatibility of fluorinated and hydrogenated segments can also be observed for PUs with bisphenol spacers (samples R_3_/100 and R_4_/100). An increase in the rigidity of the components leads to a strong increase in the sizes of the rigid domains and spacing between them (the relationship between these parameters is the same as for the systems describe above). However, the presence of trifluoromethyl groups leads to the formation of smaller rigid domains, as compared with its methyl based analog. Hence, the rigidity of components and, therefore, the molecular mobility in the system, is controlled by the morphology of PUs.

### 3.3. D WISE NMR Experiments. Correlation between Mobility and Domain Structure

A two-dimensional proton wide-line separation (2D^13^C-^1^H WISE) NMR technique provides information for the qualitative characterization of heterogeneous dynamics of various structural units in microphase-separated structures. It allows one to establish a correlation between domain structure and segmental mobility [38], making it particularly suitable for the present study. Figure 5 displays the 2D WISE spectrum as well as ^1^H slices of the sample **R_1_/100**. All the ^13^C environments have corresponding ^1^H components with narrow line widths equal to ~1 kHz. The line widths for the aromatic components (1.3 kHz) are slightly larger than for the ethylene oxide fragments (1.1 kHz). Hence, the 2D WISE method allows one to monitor only the mobile phase of the samples. The resonances that shows a widened ^1^H line width in the WISE NMR spectrum appear only for the **R_1_/50/R_3_/50** sample, which contained 50 mol.% of bisphenol A as a chain extender (Figure 5). However, the broad lines with widths equal to ~35 kHz have only the ethylene oxide component. Hence, these fragments form the most immobile fractions in the polymer sample. Their contribution increases with the increase of the bisphenol A fraction. The WISE NMR spectrum of the sample without SS blocks (Figure 5) contains, not only broad, but also narrow lines (line widths equal to ~1 kHz). Interestingly, these belong, not only to methyl groups of bisphenol A, but also to methylene groups of isocyanate components. Hence, the distribution of molecular mobilities depends, not only upon the morphology of the sample, but also on its microstructure. The WISE NMR spectrum of the sample with perfluorinated SS (Figure 6) contains only narrow lines with the strongest resonance observed from the terminal methylene groups of SS blocks. As WISE is not a quantitative technique [39,40], in order to determine the exact distribution of molecular dynamics in the systems under study, it is necessary to use the transverse magnetization relaxation time, T_2_, which provides information about molecular mobility in hard and soft domains. It also allows one to quantitatively determine a part of the different molecular motion.

#### 3.3.1. ^1^H Transverse Magnetization Relaxation

Relaxation data contain a wealth of information, but deconvolution of specific contributions can be difficult. The general principle of fitting is that it should faithfully reproduce the major features of the FID using as few adjustable parameters as possible. Based on the NMR experiments described above, we assume that the PUs under study contain rigid domains, and the soft matrix consists of two components, mobile and rigid, as commonly accepted in the study of PUs [41,42]. The relaxation decays of PUs were found to be well approximated using a linear combination of two fast-decaying Gaussian functions (for rigid domains and rigid amorphous phase) and slow-decaying single-exponential function (for mobile soft matrix):(1)A(t)=A1exp[−(tT21)2]+A2exp[−(tT22)2]+A3exp[−(tT23)]

Experimental data and fits resulting from this procedure are shown in Figure 7. Good fitting quality is seen in Figure 7, as the curves (fits) mostly pass through the dots (experimental data), and the adjustable R-square of fitting is better than 0.999 for all curves. This allows us to interpret the results qualitatively. A complete list of the fitting parameters, including the fast- and slow-decaying fractions, is displayed in Table 3, where the fractions were normalized to 100%. We start by analyzing the results obtained with Sample 1. As described above (Section 3.2), the mobile phase of this sample contains only ethylene oxide units and the less mobile phase contains both these and aromatic units. The mass fraction of the ethylene oxide units is 0.75 and, in agreement with this value, the fractions of the mobile, rigid amorphous, and rigid domains are 0.45, 0.27, and 0.27, respectively. Therefore, about half of the ethylene oxide fragments constitute the rigid phases. Similar results were obtained for perfluorinated analogues. In this sample, the SS blocks formed a mobile phase. The protons are located only in the terminal groups. As a result, it partly forms a rigid amorphous phase or interphase, as well as rigid domains. The fraction of such units is approximately one third of the total amount.

The introduction of an aromatic chain extender (bisphenol A) leads to the disappearance of the true soft mobile phase, such as that present in the sample R_1_/50/R_3_/50, and formation of the phase with restricted mobility. The presence of a kink in the aromatic chain extender makes formation of highly ordered phases less favorable. The presence of dimethylmethane or methylene units results in a higher mobility, which increases the observed relaxation times. Therefore, the phenomena in samples where zero or a low amount of SS blocks the mobility of the soft phases is related to these units, which was also observed in WISE NMR experiments.

As one can see from Table 3, the fraction of mobility connected with rigid domains (A_1_) is approximately the same for all samples under study, with one exception (**R_4_/100**). This supports our previous conclusion [26] that formation of rigid domains is governed by the presence and molar fraction of urethane bonds (molar fraction of isocyanate components) and depends relatively weakly on other factors, such as the structure of polyol components and of the chain extender. The effect of thermodynamic parameters manifests itself in the fact that, in the case of equal parts of polyol components (the formation of eutectic mixture), a more ordered polymer structure is formed, with a higher fraction of rigid domains and rigid amorphous phases. To elucidate the origin of different molecular mobility in the amorphous phase, we used dielectric spectroscopy.

#### 3.3.2. Dielectric Measurements

We used dielectric spectroscopy to directly measure the effect of PU structure on the polymer relaxation corresponding to different lengths and time scales. Modern dielectric methods cover a broad frequency range, allowing measurements of different relaxation processes simultaneously, and even the entire chain relaxation processes under favorable circumstances [43]. Dielectric spectroscopy allows one to investigate molecular dynamics in a soft matrix. However, the diffuse character of rigid domains and rigid amorphous phases provides an opportunity to observe some modes in these phases too. PUs have been thoroughly investigated using dielectric spectroscopy [44,45,46]. Four relaxations are observed in the dielectric loss spectra of all PUs: local glassy state motions (γ and β), segmental motion of the soft phase (α), and Maxwell–Wagner–Sillars interfacial polarization (MWS), together with conductivity. The latter process can be used to characterize the rigid domains because it is related to their dimensions. However, this process is often disguised by conductivity or is not observed at all as a result of diffuse domain interfaces.

The molecular mechanisms of local processes in dielectric studies often have a speculative character, because it is impossible to identify the molecular fragments involved in these processes without using other spectroscopic methods in addition. A combination of NMR methods used in our study allows one to solve this problem.

As in our previous study [36], we used complex electric modulus formalism for the analysis of dielectric data. All relaxation processes are visible as peaks in the imaginary part M’’ of the complex dielectric modulus M* (see Figure 8 and Appendix A). In order to evaluate the individual relaxation processes and their parameters quantitatively, a model function has been fitted to the dielectric data using the Havriliak–Negami phenomenological relation [47]. The fitting procedure is complicated because of the presence of incomplete peaks, in spite of the frequency window extending over more than eight decades. The quality of the fit is quite good, and the characteristic relaxation time for each relaxation process can be extracted. The main characteristic of each relaxation process is the most probable relaxation time, τ_max_ [48].

A 3D plot of M” versus frequency and temperature for the sample **R_1_/100** is shown in Figure 8. The profile of this dependence at the selected frequency clearly shows the presence of four relaxation processes. The dependences of −log τ_max_ on the inverse temperature are linear for the three low temperature peaks (Figure 9). These temperature dependences can be modeled by an Arrhenius type expression [43], and their parameters are given in Table 4. These peaks are located in the temperature/frequency region of the γ and β processes. The γ and β processes are observed in the spectra of all PUs, and are assumed to arise from the local relaxations observed in the mobile glassy phase, which are associated with the crankshaft motions of the ether oxygen containing fragments [49]. However, dielectric spectrum of this sample contains an additional relaxation process at a higher temperature (β_2_). Hence, this indicates the presence of two amorphous phases (rigid and soft), because the activation energy of β_2_ relaxation is much higher than β_1_. If we analyze the NMR spectrum of the mobile phase of this PU (Figure 1), one can see that it contains signals of methylene groups of oligomeric MDI fragments. Hence, the observed relaxation process should be related to rotations of methylene groups between aromatic units. The high value of the activation energy of this process (at least two times higher than those observed for other local processes (Table 4)) indicates its restricted character and suggests that this molecular fragment can be located in the rigid amorphous phase.

The same process is observed for the samples with perfluorinated SS blocks, and their parameters are very similar to those for the sample **R_1_/100**. Hence, it is associated with the motions in oligomeric MDI fragments. The activation energies of local processes located in perfluorinated SS are much higher in comparison with their hydrocarbon analogues. The explanation of this observation is very simple, as the rotation barriers in fluorocarbons are much higher than in their hydrocarbon analogues [27]. The β-relaxation is a doublet because perfluorinated SS blocks contain two types of fragments: -CF_2_CF_2_O- and -CF_2_O-, each of them giving rise to a separate β-relaxation. (Figure 10).

The introduction of an aromatic chain extender in the PUs under study did not change the general picture of local relaxations. A restriction of local motions related to an increase of the rigidity of the soft matrix leads to an increase of the activation energy of local relaxations (Table 4) for PUs with perfluorinated SS. However, free rotation of methyl groups around the central carbon in the bisphenol A fragment introduces disorder and leads to a decrease in the activation energy of local relaxations in comparison to the mono ethylene oxide PU (Table 4).

The segmental mobility in the PUs under study (α-relaxation) (Figure 10) correlates well with their structure. The introduction of the aromatic chain extender leads to an increase of the glass transition temperature as a result of the higher rigidity of the amorphous phase. The glass transition temperature correlates well with the ordering of the amorphous phase. The microphases, which are well separated in PU with perfluorinated SS blocks, have a maximum glass transition temperature (Table 4). The introduction of disorder into the PU structure by insertion of chain extenders that are partially included in the soft matrix leads to a decrease of the glass transition temperature (Table 3). The values of the activation energy correlate well with the parameter D (fragility), which reflects cooperativity α-relaxation [50]. Hence, the data obtained by dielectric spectroscopy are in good agreement with the data of NMR studies. These data allow us to understand the effect of the presence of two amorphous (rigid and soft) “phases” in the PU soft matrix. The local character of double relaxations (β and γ) indicates that these “phases” are related to H-bonded and free groups of soft segments. These results reveal how the mobility distribution in the soft phase of PUs is determined by the formation of H-bonded networks [51,52].

## 4. Conclusions

In summary, a variety of solid-state NMR techniques, in combination with dielectric spectroscopy, have been used to study the distinct phase structures and molecular mobilities of PUs based on oligomeric MDI, aromatic chain extenders, and ethylene oxide polyol or their perfluorinated analog as soft blocks. It is shown that, in contrast to the common model suggested by Cooper and Tobolsky in 1966, the rigid domains of microphase separated PUs are formed, not only by units containing urethane bonds produced from isocyanate components or chain extender, but also by oxyethylene fragments that form a common rigid phase. The urethane bonds and oxyethylene fragments are incorporated into both rigid and soft phases. Good agreement with the Cooper and Tobolsky model is observed only when the solubility parameters are significantly different for the hard and soft segments, as is the case of hydrocarbon aromatics and perfluoroaliphatic blocks. A complex structure of the soft polymer matrix (the presence of rigid and mobile amorphous phases) is supported by both NMR and dielectric spectroscopy. The local character of double relaxations (β and γ) indicates that rigid and mobile amorphous phases are related to H-bonded and free groups of soft segments [53,54].

## Data Availability

The data presented in this study are available on request from the corresponding author.

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
