# Peer review of "Interplay of Structural Factors in Formation of Microphase-Separated or Microphase-Mixed Structures of Polyurethanes Revealed by Solid-State NMR and Dielectric Spectroscopy"

_polymers, 2021, doi:10.3390/polym13121967_

Round 1

Reviewer 1 Report

The authors present solid-state NMR and dielectric spectroscopy studies on different polyurethane (PU) samples containing oligomeric MDI, bisphenol A and PEG 600 (and their fluorinated derivatives) that were used as soft blocks. Eleven samples were prepared for the measurements with different ratio of the ingredients. It was found based on the measurements that the ethylene oxide chain and the urethane bond are incorporated into both soft and rigid phases. In my opinion the manuscript is acceptable for publication.

My remarks, questions:

In materials the Mn of the MDi is 325, however in scheme 1 it is 340, correct it.

The authors state the rigid phase did not contain terminal ethylene oxide units, however a peak with minor intensity can be observed in Fig 1 (a1). This part should be rephrased.

Fig 1b is not mentioned in the text.

In some cases the “Pus” should be modified to PUs. For instance lines 266, 392.

In the manuscript the “as one can see” phrase is repeated too many times. It should be replaced.

The ratio of the hard soft domains in the PU affects on the physical properties. What kinds of measurements (for example tensile test) were performed with the samples that can confirm the NMR results in the practice?

Author Response

Response to reviewer 1.

We made next corrections

  1. In exp.part change the Mn for oligomeric MDI to 340
  2. Including in text reference to Fig.1b
  3. Rephrasing sentence about Fig.1a
  4. It is important to note that the rigid phase with restricted mobility practically did not contain terminal fragments of ethylene oxide units, and, as seen from the spectra of the R1/100-sample normalized with respect to the signal intensity of the ethylene oxide (the integral intensity of this line is the same in all spectra) (Fig.1a),
  5. We delete ‘As one can see’ in mostly sentences
  6. The ratio of the hard soft domains in the PU affects on the physical properties. What kinds of measurements (for example tensile test) were performed with the samples that can confirm the NMR results in the practice?

The influence this ration on mechanical properties of PUs is well known. For this samples we did not made any mechanical tests but for similar compositions you can see our paper

  1. Pikhurov, D.V.; Sakhatskii, A.S.; Zuev, V.V. Rigid polyurethane foams with infused hydrophilic/ hydrophobic nanoparticles: Relationship between cellular structure and physical properties. Eur.Polym. J. 2018, 99, 403-414.

We use IR spectroscopy for estimation of this ratio

VVZuev

Reviewer 2 Report

The manuscript “Interplay of structural factors in formation of microphase-separated or microphase-mixed structures of polyurethanes revealed by solid-state NMR and dielectric spectroscopy” describes an extensive study of a large number of polyurethanes samples. Many different solid state NMR experiments, exploring both high and low resolution, are reported together with DS measurements. All the data are well described and the results are clearly presented.

In my opinion, there are two major points to be clarified:

  • The authors refer to microphase in the title and throughout the manuscript, but spin diffusion experiments and SEM images indicate that the smallest dimensions of different domains are lower than 28 nm in all the samples. I would suggest revising the term micro- and using nano-phase o nano-domains instead, or comment on this.
  • 1H T2 were measured by Hahn Echo pulse sequence. Since this sequence only refocuses chemical shift and J coupling interactions it is not the best choice for the very rigid domains, where homonuclear dipolar interaction dominates. For the evaluation of the rigid fraction sequences like solid echo or magic sandwich echo are indicated. Can the author comment this choice? In the SI solid echo is reported only for one sample but the scope of this figure is not clear. Please specify if this support the choice of Hahn echo for FID analysis. Still, a comparison between the two kinds of echoes (Hanh and solid) could be useful.

Other minor issues are:

  • λDPS values are reported in table 1 but the definition of this parameter is only in the SI. It would be easier for the reader to have it in the main text or in the caption of table 1.
  • Please verify the phrase at lines 250-251: “the relative fraction each varying significantly vary from sample to sample”
  • Please reword caption of Fig 6. This is a 3D view of 2D spectra, not 3D NMR spectra.
  • In the SI pag. 7, the reference at the paper n°2 seems wrong to me. The correct reference for CPSP-MAS experiments is 1. Please check.
  • Please add page numbers and number all the figures in the SI.

Author Response

Response to reviewer 2.

We made next corrections

  1. We accept all minor suggestions and rewrite sentences, caption of Figure 6, and made changes in Supplementary
  2. I am agree with comment about terminology microphase/ nanophase separation. However. in the field of PUs the ‘microphase’ is common term although all known about nanosized dimension of rigid domains.
  3. The definition of λDPS is given in Table 1.
  4. According the measurements of relaxation times.

T2 determination for such heterogeneous materials is challenging. We use the Hahn-echo sequence to describe more accurately the behavior of the more rapidly relaxing components. The differences in signal intensity between the Hahn/solid echo and the single-90 FID for both the rigid and the mobile components were less than 2%.(see supporting)
